# Comparison of Ferroptosis-Inhibitory Mechanisms between Ferrostatin-1 and Dietary Stilbenes (Piceatannol and Astringin)

**DOI:** 10.3390/molecules26041092

**Published:** 2021-02-19

**Authors:** Ban Chen, Xican Li, Xiaojian Ouyang, Jie Liu, Yangping Liu, Dongfeng Chen

**Affiliations:** 1School of Chinese Herbal Medicine, Guangzhou University of Chinese Medicine, Guangzhou 510006, China; imchenban@foxmail.com (B.C.); oyxiaojian55@163.com (X.O.); 2School of Basic Medical Science, Guangzhou University of Chinese Medicine, Guangzhou 510006, China; 15014173165@163.com; 3The Research Center of Basic Integrative Medicine, Guangzhou University of Chinese Medicine, Guangzhou 510006, China; 4The Fourth Clinical Medical College, Guangzhou University of Chinese Medicine, Waihuan East Road No. 232, Guangzhou Higher Education Mega Center, Guangzhou 510006, China; 20182119168@stu.gzucm.edu.cn

**Keywords:** ferroptosis, Ferrostatin-1, piceatannol, astringin, electron transfer, hydrogen-donating, stilbene, quantum chemistry

## Abstract

Synthetic arylamines and dietary phytophenolics could inhibit ferroptosis, a recently discovered regulated cell death process. However, no study indicates whether their inhibitory mechanisms are inherently different. Herein, the ferroptosis-inhibitory mechanisms of selected ferrostatin-1 (Fer-1) and two dietary stilbenes (piceatannol and astringin) were compared. Cellular assays suggested that the ferroptosis-inhibitory and electron-transfer potential levels decreased as follows: Fer-1 >> piceatannol > astringin; however, the hydrogen-donating potential had an order different from that observed by the antioxidant experiments and quantum chemistry calculations. Quantum calculations suggested that Fer-1 has a much lower ionization potential than the two stilbenes, and the aromatic N-atoms were surrounded by the largest electron clouds. By comparison, the C4′O-H groups in the two stilbenes exhibited the lowest bond disassociation enthalpies. Finally, the three were found to produce corresponding dimer peaks through ultra-performance liquid chromatography coupled with electrospray ionization quadrupole time-of-flight tandem mass spectrometry analysis. In conclusion, Fer-1 mainly depends on the electron transfer of aromatic N-atoms to construct a redox recycle. However, piceatannol and astringin preferentially donate hydrogen atoms at the 4′-OH position to mediate the conventional antioxidant mechanism that inhibits ferroptosis, and to ultimately form dimers. These results suggest that dietary phytophenols may be safer ferroptosis inhibitors for balancing normal and ferroptotic cells than arylamines with high electron-transfer potential.

## 1. Introduction

Ferroptosis is a recently discovered non-apoptotic form of regulated cell death [1], and ferroptosis inhibition is considered a novel cytoprotective strategy [2]. Recently, scientists have synthesized several arylamine ferroptosis inhibitors, such as ferrostatin-1 (Fer-1) [3,4], which mediates catalytic recycling to effectively inhibit ferroptosis [3]. Notably, these synthesized arylamine inhibitors have not been consumed daily and may have some certain cytotoxic risks if used for cytoprotective strategies [5].

Scientists have searched for ferroptosis inhibitors among dietary phytophenols. For example, our team found that two dietary phytophenols (corilagin and 1,3,6-tri-*O*-galloyl-*β*-d-glucopyranose) could act as ferroptosis inhibitors [6]. The safety of dietary ferroptosis inhibitors is supported by a basic fact, namely, human beings have consumed these phytophenols for a long history. Parent plants even play a beneficial role in cellular viability in traditional Chinese medicine. These beneficial effects mean that dietary phytophenols can maintain the balance between living cells and ferroptotic cells. However, it is unclear whether there is an inherent difference between the synthesized arylamine and dietary phytophenols in their ferroptosis-inhibitory mechanisms. To date, there have been no comparative studies between them.

In this study, we investigated the difference in ferroptosis-inhibitory mechanisms between Fer-1 and two dietary phytophenols: piceatannol and its glucoside astringin (i.e., piceatannol-3-*O*-*β*-d-glucopyranoside; Figure 1). The two phytophenols are stilbenes that co-exist in grapes, a widely consumed fruit [7,8]. Their structural similarity offers the possibility for detailed structure–activity relationship analysis.

Fer-1 and the two stilbenes were evaluated using three cellular assays, i.e., 4,4-difluoro-5-(4-phenyl-1,3-butadienyl)-4-bora-3a,4a-diaza-s-indacene-3-undecanoic acid (C11-BODIPY) assay, cell counting kit-8 kit (CCK-8) assay, and flow cytometry assay. To emphasize the therapeutic relevance, bone marrow-derived mesenchymal stem cells (bmMSCs), which are important seed cells for stem cell transplantation engineering [9,10], were utilized as the cell model in this study. The ferroptosis-inhibitory mechanisms were subsequently explored through chemical approaches, including cutting-edge ultra-performance liquid chromatography coupled with electrospray ionization quadrupole time-of-flight tandem mass spectrometry (UPLC-ESI-Q-TOF-MS/MS) and quantum chemistry calculations based on M06-2X-D3/6–311+G** and M06-2X-D3/def2-TZVPD levels [11,12,13,14,15,16,17,18,19,20]. High-accuracy mass spectra can provide reliable evidence, and quantum chemistry calculations based on M06-2X-D3/6–311+G** and M06-2X-D3/def2-TZVPD levels can offer detailed information for mechanism elucidation.

## 2. Results and Discussion

In the present study, the ferroptotic bmMSCs model was established using erastin, an imidazole ketone inducer [21,22]. Erastin inhibits the biosynthesis of glutathione peroxi-dase 4 (GPX4), a cellular antioxidant enzyme. The decrease in GPX4 further causes increases in various reactive oxygen species (ROS), including the HO^•^ radical, phospholipid lipid peroxide (L-OOH), the lipid hydroperoxyl radical (L-OO^•^), the phospholipid alkoxyl radical (PL-O^•^), and the phospholipid hydroperoxyl radical (PL-OO^•^). All of these species oxidatively damage lipid bilayers in the cell membrane, which ultimately leads to cellular death [3,23,24,25].

The erastin-treated bmMSCs acted as the model group, while medium-treated bmMSCs functioned as the control group. The two groups were measured using a C11-BODIPY lipid peroxidation (LPO) sensor [26]. As shown in Figure 2A, the model group showed much darker green than the control group, which implied that erastin had already induced LPO accumulation in the bmMSCs. Correspondingly, the model group also showed low cell viabilities in CCK-8 (51.4%, Figure 2B) and flow cytometric assays (51.9%, Figure 2C), while the control group showed high cell viabilities in the two assays (100% and 98.7%, respectively).

The fluorescence intensity and cell viabilities of three samples (Fer-1, piceatannol, and astringin) were similar and differed from the model and control groups (Figure 2), which indicates that the three samples effectively inhibited LPO accumulation and sub-sequent ferroptosis and thus improved the viability of bmMSCs treated with erastin. For example, the 100 μM piceatannol, 100 μM astringin group, and 1 μM Fer-1 groups exhibited 85.9%, 77.3%, and 99.8 % cell viabilities, respectively, in the CCK-8 assay (Figure 2B).

The ferroptosis-inhibitory levels decreased in the order of Fer-1 >> piceatannol > as-tringin. This order greatly differed from that for ^•^O_2_^−^-scavenging, in which Fer-1 exhibited the lowest ROS-scavenging (antioxidant) level (Table 1). The inactivity of Fer-1 for ^•^O_2_^−^-scavenging can explain a previous finding that Fer-1 exhibited ferroptosis inhibition via catalytic recycling rather than ROS scavenging [3].

The activities of piceatannol and astringin for superoxide anion radical (^•^O_2_^−^) scavenging suggest that they likely mediate ROS scavenging, a conventional antioxidant mechanism, to inhibit ferroptosis. This proposal is further supported by the following three facts: (1) Ferroptosis is closely associated with extensive ROS (including LPO) production [21,27,28,29]; (2) the relative ferroptosis-inhibitory levels correlate with the relative antioxidant levels of piceatannol and astringin (Table 1 and Figure 2); however, the antioxidant difference between the two stilbenes can be attributed to 3-OH glycosylation, according to a previous antioxidant structure–activity relationship analysis [30,31]; and (3) nowadays, many natural phenolic antioxidants have already been proven to be ferroptosis inhibitors, such as corilagin [6], baicalein [32], and isorhapontigenin [33].

In addition to the ^•^O_2_^−^-scavenging assay, a PTIO^•^-scavenging assay was used to characterize the antioxidant levels in aqueous solution. According to the IC_50_ values in Table 1, the two stilbenes displayed much higher antioxidant levels than Fer-1. However, the PTIO^•^-scavenging assay has been recently reported to represent a hydrogen-donating process [34]; thus, the results suggest that the two stilbenes present hydrogen-donating potential. This was confirmed by the UPLC-ESI-Q-TOF-MS/MS analysis.

As shown in Figure 3A, PTIO^•^-treated piceatannol yielded a chromatographic peak at 1.497 min. The peak, however, afforded an *m/z* 486.1283 molecular ion peak in the MS spectra (Figure 3B). The value represented the loss of two H atoms as compared to the double of the molecular weights of piceatannol (*m/z* 244.0736). There were only 7 × 10^−6^ relative derivations, suggesting that one stilbene molecule indeed donated one hydrogen atom. Similar results were also observed for PTIO^•^-treated astringin (Figure 3C–E) and PTIO^•^-treated Fer-1 (Figure 3F,G).

Quantum chemistry calculations indicated that two stilbenes showed the lowest bond dissociation enthalpies (BDEs) values at the corresponding 4′-OH (Figure 4), and the 4′-OH was hence considered to be the dominant hydrogen-donating position in aqueous solution [35]. In terms of the fact that low BDE values represent high hydrogen-donating potential [15] and the lowest BDE values in Table 2, the hydrogen-donating potential decreased in the order: piceatannol > astringin >> Fer-1. The two stilbenes may donate hydrogen at 4-OH to scavenge ROS to further inhibit ferroptosis. Ultimately, they were converted into corresponding dimers.

On the contrary, electron transfer was strong for Fer-1. As seen in Table 1, Fer-1 gave much lower IC_50_ values than the two stilbenes in the Fe^3+^-reducing experiment, which is an electron-transfer-based antioxidant assay. Such an advantage of Fer-1 was further supported by quantum chemistry calculations. Notably, the ionization potential (IP) values of Fer-1 were calculated to be much lower than those of the two stilbenes in aqueous and methanol solutions (Figure 4 and Figure 5), which indicates that Fer-1 possesses high electron-transfer potential under both lipophilic and aqueous environments [35]. The electron-transfer advantage facilitates Fer-1 to mediate electron-transfer and exert ferroptosis-inhibitory action in the lipid bilayer of the cellular membrane.

To better understand the electron-transfer advantage of Fer-1, the electron density difference between the neutral molecule and cationic radical was further calculated. As seen in the isosurface maps (Figure 4 and Figure 5), there were large electron clouds around the C_1_-N and C_2_-N, which indicate that Fer-1 may undergo electron transfer at the aromatic N-atoms in both aqueous and methanol solutions. This further supports the hypothesis that Fer-1 exhibits ferroptosis-inhibiting activity mainly via a redox-based catalytic recycle [3], and goes against radical trapping, a hydrogen-donating process, which is the main approach for ferroptosis inhibition [5,29].

It is clear that the ferroptosis-inhibitory mechanisms between Fer-1 and dietary stilbenes are quite different. Fer-1 has been reported to mediate catalytic recycling, a redox reaction combined with metal binding, to inhibit ferroptosis [3]. Our study further revealed that its redox potential is associated with its strong electron-transfer ability. The redox reaction can circularly occur only between the iron ion and the Fer-1 molecule. Thus, there is no isolated Fe^2+^ to catalyze the Fenton-like reaction to produce ROS (e.g., ^•^OH), which can effectively prevent oxidative damage in cells. This can be responsible for the effectiveness of the cellular assays (Figure 2). However, the interaction between Fer-1 and ROS was inevitable; in this case, Fer-1 could also donate hydrogen to PL-O^•^ [3], and finally, form a dimer.

It should be noted that arylamine Fer-1 has been documented to be cytotoxic (TC_50_ 126 μM) [5], and its cytotoxicity may be associated with low IP values. Structurally, the exocyclic N-atom of Fer-1 is similar to those of cytosine, adenine, and guanine, three human nucleobases. Nevertheless, the three bases present much higher IP values (124.3, 117.4, and 107.6 kcal/mol, Figure 6) than Fer-1. In fact, there is no similar aromatic exocyclic N-atom in human cells, except for those of three base pairs. This suggests that Fer-1, with a high electron-transfer potential, is actually absent in human cells. Similar arylamine antioxidants have been used only in industrial additives [36]. Moreover, Fer-1 was found to be slightly soluble in aqueous solution in our experiment (Table 1).

Two dietary stilbenes, piceatannol and astringin, are compatible in human cells because they have been widely consumed by humans for a long time. The ferroptosis-inhibitory mechanism of the two dietary stilbenes has been demonstrated to involve an antioxidant process, during which they undergo hydrogen donation to scavenge ROS and inhibit ferroptosis. The hydrogen donation first occurs at 4′-OH and finally results in a dimer as well. Of course, the detailed structure of the dimer requires work in the future because of the variety of linking sites.

The findings based on piceatannol and astringin can be generalized to the stilbene family. This is because the antioxidant bioactivity of all phenolic stilbenes can be attributed to the same functional group, i.e., phenolic -OH. This deduction can predict that dietary stilbenes may be a resource of exogenous and natural ferroptosis inhibitors and usually mediate antioxidant mechanisms to maintain the balance between ferroptotic and normal cells. Supplementation of dietary stilbenes may be a safe and available approach for ferroptosis inhibition.

## 3. Materials and Methods

### 3.1. Animals, Biological Kits and Chemicals

Four-week-old male Sprague-Dawley (SD) rats were supplied by the Experimental Animal Center of Guangzhou University of Chinese medicine (Guangzhou, China). The complete glucose medium was obtained from Cyagen Biosciences (Guangzhou, China); Dulbecco’s modified eagle medium (DMEM), fetal bovine serum (FBS), and trypsin were obtained from Molecular Probes (Carlsbad, CA, USA); CCK-8 was from Dojindo Chemistry Research Institute (Kumamoto, Japan); the Annexin V/propidium iodide (PI) assay kit was obtained from BD Biosciences (Carlsbad, CA, USA). C11-BODIPY was obtained from Molecular Probes (CA, USA); and erastin was obtained from MedChemExpress (Monmouth Junction, NJ, USA).

Piceatannol (Appendix A) was purchased from Biopurify Phytochemicals Co., Ltd. (Chengdu, China); astringin (piceatannol-3-*O*-*β*-d-glucopyranoside, Appendix A) was purchased from ChemFaces (Wuhan, China); Ferrostatin-1 was from Glpbio (Shanghai, China); pyrogallol, 2,4,6-tri(2-pyridyl)-s-triazine (TPTZ) and Trolox were obtained from Sigma-Aldrich (Shanghai, China); and PTIO radical was from TCI Chemical Co., Ltd. (Shanghai, China). Water, methanol, and formic acid were of HPLC grade; FeCl_3_·6H_2_O and other reagents of analytical grade were purchased from Guangdong Guanghua Chemical Factory (Shantou, China).

### 3.2. Protection of Erastin-Induced Ferroptosis in bmMSCs

The animal experiment was approved by the Institutional Animal Ethics Committee in Guangzhou University of Chinese (approval number 20180034) and modified appropriately [37]. In short, eight four-week-old SD rats (60–80g) were euthanized by cervical dislocation to obtain the bone marrow of the femur and tibia. The bone marrow samples were diluted with low-glucose DMEM containing 10% FBS, and bmMSCs were prepared by gradient centrifugation at 900 g for 30 min on 1.073 g/mL Percoll. The prepared cells were detached by treatment with 0.25% trypsin and passaged into cultural flasks at 1 × 10^4^/cm^2^ for cultured cell homogeneity.

To measure the inhibitory bioactivities of Fer-1, piceatannol, and astringin toward erastin-induced ferroptosis in bmMSCs, three assays were employed in the study, including the C11-BODIPY assay, CCK-8 assay, and flow cytometric assay. The bmMSCs prepared above were divided into three groups (i.e., control group, model group, and sample group) for comparison in each assay [38].

In the C11-BODIPY assay, the bmMSCs were seeded into 12-well plates at 1 × 10^6^ cells per well and were adhered for 24 h [39]. In the control group, the adhered cells were incubated in Stel Basal medium for 12 h; in the model and sample groups, the cells were incubated in the presence of erastin (20 μM). After the medium was removed, the cells in the control and model groups were incubated in Stel Basal medium for 12 h, while the cells in the sample group were incubated with different samples (3 μg/mL). Finally, the cells of three groups were analyzed by a C11-BODIPY sensor (2.5 μM). The images were taken using a fluorescence microscope.

In the CCK-8 assay, the bmMSCs were seeded at 1×10^6^ cells per well into 96-well plates [40]. After adherence for 12 h, the incubated cells were treated as above by adding 10 μL CCK-8, and the culture was incubated for an additional 3 h. After the culture medium was discarded, the absorbance was measured at 450 nm on a Bio-Kinetics reader. According to the *A*_450 nm_ values, the survival was calculated. Each sample test was repeated in three independent wells.

The cultured bmMSCs were also seeded at 1 × 10^6^ cells per well into 96-well plates in the flow cytometric assay [41]. They were washed twice with cold PBS, and cells were resuspended in 1 × Binding buffer at a concentration of 1 × 10^6^ cells/mL. Then, 100 μL of the solution (1 × 10^5^ cells) were transferred to a 5-mL culture tube, and 5 μL of FITC Annexin V and 5 μL PI were added to the culture tube. The cells were vortexed gently and were incubated for 15 min at room temperature in the dark; finally, 400 mL of 1 × Binding Buffer were added to each tube after adherence for 12 h. The three groups were analyzed by flow cytometry within 1 h. Each sample test was repeated in three independent wells.

### 3.3. Antioxidant Colorimetric Assays

The antioxidant colorimetric assays in the present study included the (^•^O_2_^−^)-scavenging assay, Fe^3+^-reducing assay, and PTIO^•^-scavenging assay. In the three assays, Fer-1, piceatannol, and astringin acted as samples, while Trolox acted as the positive control. Fer-1, piceatannol, astringin, and Trolox were dissolved in methanol.

The ^•^O_2_^−^-scavenging potential was determined using pyrogallol autoxidation with some modification [42]. Tris-HCl buffer (0.05 M) was adjusted to pH 7.4 using about 1mM Na_2_EDTA, pyrogallol solution (60 mM) was prepared in 1 mM HCL, and *x* μL sample solution (specific concentration of each sample was shown in Appendix A) was mixed with (980 − *x*) μL Tris-HCl buffer and 20 μL of pyrogallol solution. The absorbance of the mixture at 325 nm was measured (Unico 2100, Unico, Shanghai, China) using Tris-HCl buffer as a blank every 30 s for 5 min to obtain the increased value (*ΔA*). The ^•^O_2_^–^ scavenging ability was calculated as Equation (1):(1)$$\mathrm{Scavenging}\;\%\;=\;\left(1-\frac{\Delta A}{{\Delta A}_{0}}\right)\times 100\%,$$
where *ΔA* is the increase of the absorbance value of the mixture with samples in 5 min, and *ΔA_0_* is that without samples.

The Fe^3+^-reducing assay is also called the ferric ion (Fe^3+^) reducing antioxidant power (FRAP) evaluation assay. It was carried out based on the method of Benzie [43] and slightly modified by a previous report [44]. In brief, the FRAP reagent was prepared by mixing 10 mM TPTZ (prepared in 40 mM HCl), 20 mM FeCl_3_·6H_2_O (prepared in water), and 0.25 M CH_3_COOH/CH_3_COONa buffer (pH 3.6) at a ratio of 1:1:10. Then, 0–10 μL methanol sample solution (0.5 mg/mL for both Fer-1, piceatannol, astringin, and Trolox, see in Appendix A) were respectively added to 10–0 μL methanol and treated with 80 µL of FRAP reagent. The mixture was vigorously shaken for 10 s and left in the dark at room temperature for 30 min. Subsequently, the absorbance was measured at 450 nm (*A_593nm_*) using FRAP reagent as the blank on a microplate reader (Multiskan FC, Thermo Scientific, Shanghai, China). The relative reducing power of the sample as compared to the maximum absorbance was calculated by Equation (2):(2)$$\mathrm{Relative}\;\mathrm{reducing}\;\mathrm{power}\;\%\;=\;\frac{A\;-A_{min}}{A_{max}\;-\;A_{min}}\times 100\%,$$
where *A_min_* is the lowest *A_593nm_* value in the experiment, *A* is the *A_593nm_* value of the mixture with samples, and *A_max_* is the greatest *A_593nm_* value in the experiment.

The PTIO^•^-scavenging assay was based on the method established by our team [45]. PTIO radical was dissolved in phosphate buffers (pH 4.5 or 7.4) to prepare a PTIO^•^ solution; *x* µL sample solution (specific concentration of each sample is shown in Appendix A) was mixed with (20 – *x*) µL phosphate buffers, and treated with 80 µL PTIO^•^ solution. After incubation in the dark at room temperature for 1 h, the absorbance of the mixture at 560 nm was measured on the microplate reader with a blank (phosphate buffers). The PTIO^•^ scavenging percentage was calculated using Equation (3):(3)$$\mathrm{Scavenging}\;\%\;=\;(1-\frac{A}{A_{0}})\times 100\%,$$
where *A_0_* is the absorbance value without samples, and *A* is the absorbance value with samples.

### 3.4. Computational Details

To ensure correlations between chemical experiments, all calculations were performed in the aqueous and methanol solvation model based on density (SMD) using the Gaussian 16 software [46]. The geometry optimization and vibration frequency of all stationary points (neutral molecules, free radicals, and cationic free radicals) were calculated using the M06-2X-D3 hybrid functional in conjunction with the 6–311+G** basis set because the M06-class functionals have been proven to yield satisfactory overall performance for thermodynamic and kinetic calculations in organic and biological systems that involve free radical reactions [47]. The optimized structures at the local minimum were ensured by the absence of an imaginary frequency. Single-point energy calculations were further performed using M06-2X-D3 combined with a larger basis set def2-TZVPD with optimized geometries. All output files were visualized and analyzed using GaussView 6.0, VMD 1.9.3, and Multiwfn 3.8 programs [48,49,50]. The calculation formula for the BDEs of C-H, N-H, O-H, and IP of each molecule are as Equations (4) and (5):(4)$$\mathrm{BDE}=H\left({\mathrm{ArO}}^{•}\right)+H\left(H^{•}\right)-H\left(\mathrm{ArOH}\right),$$
(5)$$\mathrm{Relative}\;\mathrm{reducing}\;\mathrm{power}\;\%\;=\;\frac{A\;-A_{min}}{A_{max}\;-\;A_{min}}\times 100\%,$$
where *H* is the enthalpy value at 298 K and the calculated *H*(H^•^), *H*(e^−^), and *H*(H^+^) values in water or methanol were obtained from the literature [51]. *H*(ArOH), *H*(ArO^•^), and *H*(ArOH^+•^) were calculated as the sum of the thermal correction to enthalpy and the single-point electronic energy.

### 3.5. UPLC-ESI-Q-TOF-MS Analysis of PTIO^•^ Reaction Products with Fer-1 and Stilbenes 

The methanol solution of each sample (Fer-1, piceatannol, astringin) and PTIO^•^ were mixed with a molar ratio of 1:2, and the resulting mixture was incubated for 12 h in the dark at room temperature. Subsequently, the mixture was passed through a 0.22-μm filter for UPLC-ESI-Q-TOF-MS analysis [52].

The LC-MS instrument consisted of a Shimadzu UPLC LC-30AD system and an AB Sciex 5600^+^ Triple-TOF mass spectrometer. The chromatogram was achieved on a C_18_ column (2.1 mm i.d. × 100 mm, 1.6 μm, Phenomenex Inc., Torrance, CA, USA) with a gradient elution of mobile phase A (methanol) and phase B (0.1% formic acid water) at the flow rate of 0.20 mL/min and column temperature maintained at 30 °C. The gradient elution program was as follow: 0–2 min, maintain 30% B; 2–10 min, 30–0% B; 10–12 min, 0–30% B [53].

The mass spectrometry detection was performed with negative electrospray ionization mode. The MS and MS/MS spectra were obtained in the range *m/z* 100–2000. The final optimized mass parameters were set as follows: ion spray voltage, −4500 V; ion source heater temperature, 550 °C; curtain gas pressure, 30 psi; nebulizing gas pressure, 50 psi; and Tis gas pressure, 50 psi. The delustering potential was set at −100 V, whereas the collision energy was set at −45 V with a collision energy spread of 15 V. The final products were quantified by extracting the corresponding molecular formulae from the total ion chromatogram and integrating the corresponding peak using PeakView 2.0 software (AB Sciex, Framingham, MA, USA) [54].

### 3.6. Statistical Analysis

Each anti-ferroptosis assay or antioxidant assay was performed in triplicates; all the data were statistically analyzed and expressed as mean values ± standard error of the mean. The IC_50_ value was defined as the final concentration of 50% radical scavenging (or relative reducing power) and calculated via the dose–response curves (Appendix A), which were plotted using Origin 2017 (OriginLab, Northampton, MA, USA). The significance of the data difference shown in Table 1 was analyzed by one-way analysis of variance (ANOVA) using SPSS 13.0 software (SPSS Inc., Chicago, IL, USA), with statistical significance set at *p* < 0.05.

## 4. Conclusions

Fer-1 has stronger electron transfer and weaker hydrogen-donating potential than the two dietary stilbenes, piceatannol and astringin. In ferroptosis-inhibitory action, Fer-1 mainly transfers electrons at the N-position to mediate the redox-based catalytic recycle and finally result in the formation of a Fer-1 dimer. However, the two stilbenes prefer to denote the hydrogen atom at 4′-OH to initiate a conventional antioxidant mechanism, and finally give rise to dimers as well. The findings of the two dietary stilbenes can be generalized into the dietary stilbene family. Thus, dietary stilbenes may be safer ferroptosis inhibitors to balance the normal cells and ferroptotic cells than the arylamines.

## Figures and Tables

**Figure 1 molecules-26-01092-f001:**
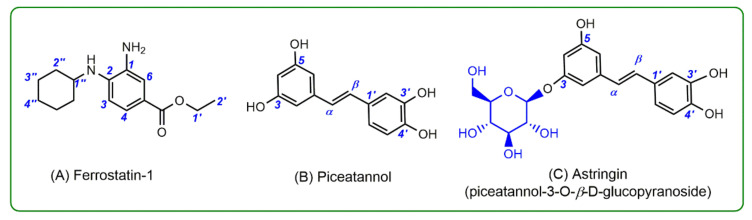
Structures of Ferrostatin-1 (Fer-1, **A**), piceatannol (**B**), and astringin (piceatannol-3-*O*-*β*-d-glucopyranoside, **C**).

**Figure 2 molecules-26-01092-f002:**
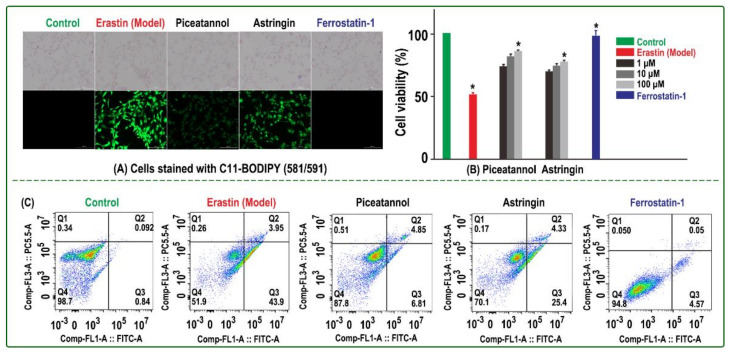
Inhibitory effect of Fer-1, piceatannol, and astringin on erastin-induced ferroptosis in bmMSCs: (**A**) C11-BODIPY assay; (**B**) CCK-8 assay; the control group was cultured in medium only, while the model group was treated with erastin. The sample group was damaged by erastin and then treated with 1, 10, and 100 μM piceatannol or astringin. The positive control group was damaged by erastin and then treated with Fer-1. Each value is expressed as the mean ± SD, *n* = 3; *, *p* < 0.05, significant difference vs the model group. (**C**) Flow cytometry assay; the assay was conducted to distinguish live cells (**Q4**), necrotic cells (**Q1**), early apoptotic cells (**Q3**), and late apoptotic cells (**Q2**). The experiment was performed with three different batches of cells, and each batch was tested in triplicate.

**Figure 3 molecules-26-01092-f003:**
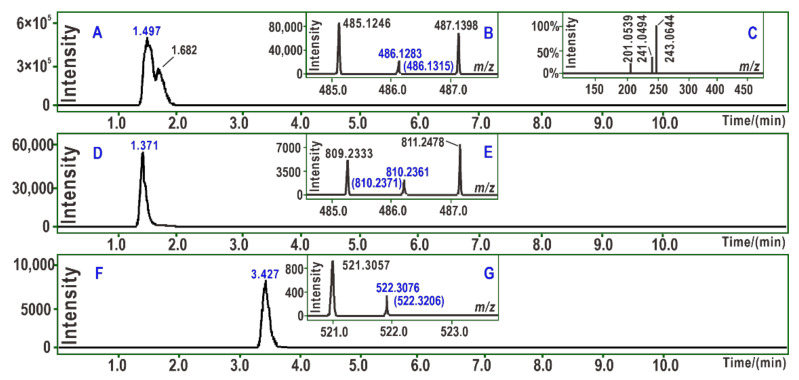
Main UPLC-ESI-Q-TOF-MS analysis results of ferroptosis-inhibitor and PTIO radical product mixtures. **A**, chromatogram of the piceatannol-piceatannol dimer product when the formula [C_28_H_22_O_8_-H]^−^ was extracted; **B**, primary MS spectra of piceatannol-piceatannol dimer product; **C**, secondary MS spectra of piceatannol-piceatannol dimer product; **D**, chromatogram of the astringin-astringin dimer product when the formula [C_40_H_42_O_18_-H]^-^ was extracted; **E**, primary MS spectra of astringin-astringin dimer product; **F**, chromatogram of the Fer-1-Fer-1 dimer product when the formula [C_30_H_42_N_4_O_4_-H]^-^ was extracted; G, primary MS spectra of Fer-1-Fer-1 dimer product. The values in brackets are the *m/z* values that were calculated based on the relative atomic masses. The *m/z* values without brackets are the experimental values obtained in the study.

**Figure 4 molecules-26-01092-f004:**
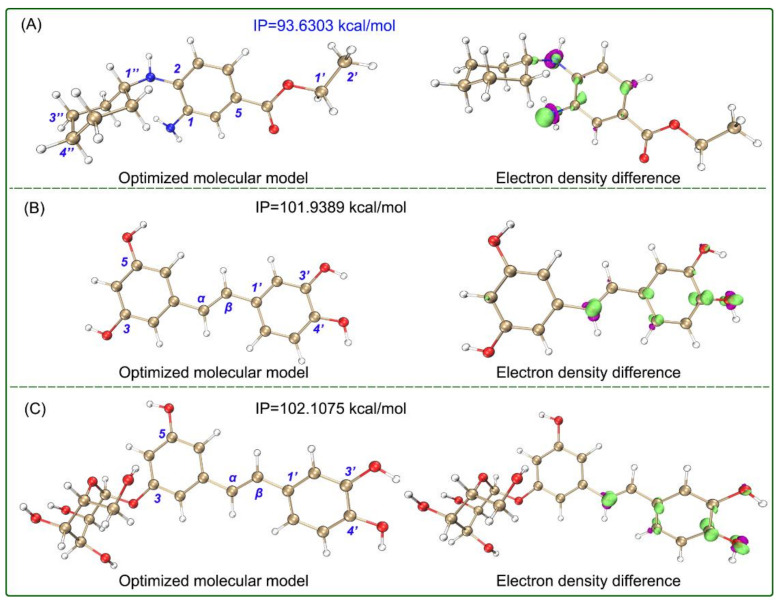
Optimized molecular model with IP value and isosurface map of electron density difference between neutral molecules and cationic radicals (**A**, Fer-1; **B**, piceatannol; **C,** astringin) in the aqueous SMD solvation model. The calculations were based on M06-2X-D3/6-311+G** and M06-2X-D3/def2-TZVPD calculation levels. In the isosurface map, lime and purple colors correspond to the regions where the electron density increased and decreased, respectively (isovalue = 0.008). The Cartesian coordinates and energies of the optimized neutral molecules are listed in Appendix A.

**Figure 5 molecules-26-01092-f005:**
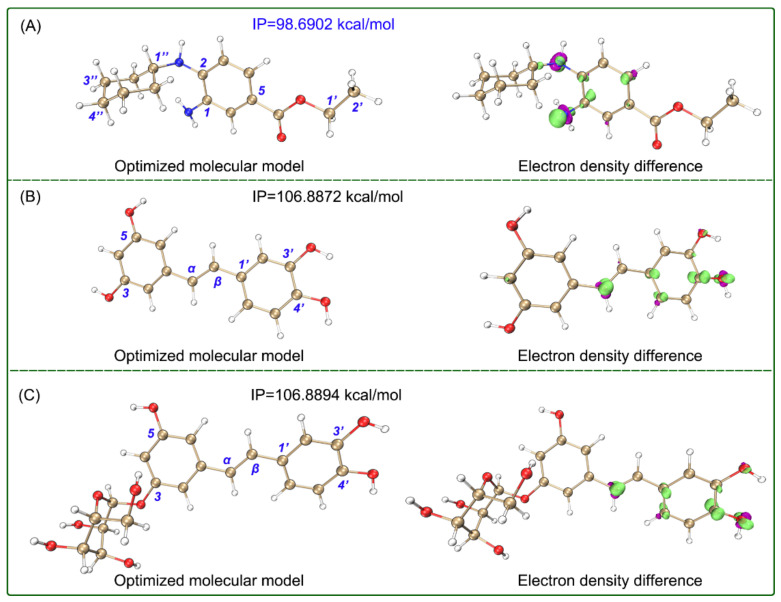
Optimized molecular model with the IP value and isosurface map of the electron density difference between a neutral molecule and cationic radical (**A**, Fer-1; **B**, piceatannol; **C,** astringin) in the methanol SMD solvation model. The calculations were based on M06-2X-D3/6-311+G** and M06-2X-D3/def2-TZVPD calculation levels. In the isosurface map, lime and purple colors correspond to the regions where the electron density is increased and decreased, respectively (isovalue = 0.008). The Cartesian coordinates and energies of the optimized neutral molecules are listed in Appendix A.

**Figure 6 molecules-26-01092-f006:**
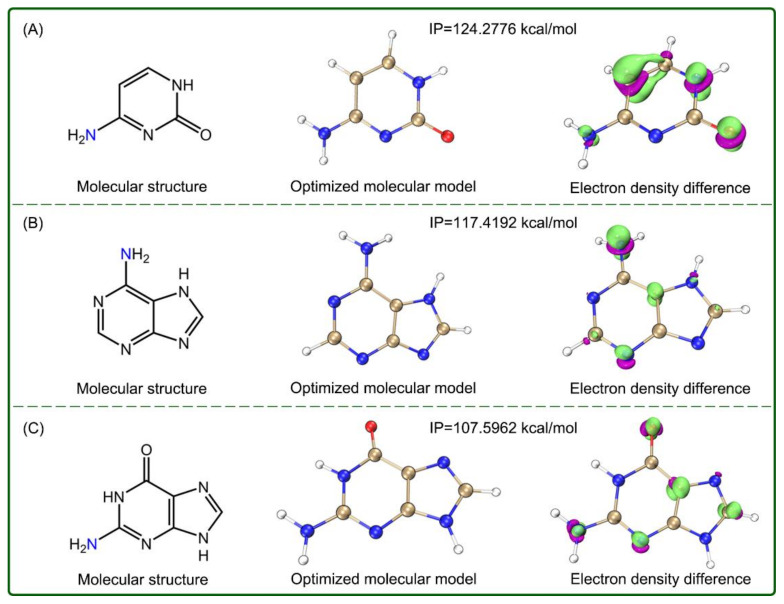
The structure, optimized molecular model along with IP value, and isosurface map of the electron density difference between the molecules and cationic radicals of three nucleobases in the aqueous SMD solvation model. **A**, cytosine; **B**, adenine; **C**, guanine. The calculations were based on M06-2X-D3/6–311+G** and M06-2X-D3/def2-TZVPD calculation levels. In the isosurface map, lime and purple colors correspond to the regions where the electron density increased and decreased, respectively (isovalue = 0.01). The Cartesian coordinates and energies of the optimized neutral molecules are listed in Appendix A.

**Table 1 molecules-26-01092-t001:** IC_50_ values (μM) of Fer-1, piceatannol, and astringin in antioxidant colorimetric assays.

	Fer-1	Piceatannol	Astringin	Trolox
^•^O_2_^−^-scavenging	58.2 ± 2.7 ^b^	30.7 ± 0.4 ^a^	37.8 ± 0.3 ^a^	127.0 ± 4.2 ^c^
PTIO^•^-scavenging (pH 4.5)	878.9 ± 33.0 ^d^	312.5 ± 12.0 ^b^	411.1 ± 25.6 ^c^	206.2 ± 8.5 ^a^
PTIO^•^-scavenging (pH 7.4)	>10,000	219.8 ± 7.0 ^a^	368.8 ± 12.9 ^b^	310.7 ± 5.8 ^b^
Fe^3+^-reducing	25.6 ± 0.3 ^a^	68.2 ± 5.3 ^b^	178.9 ± 2.8 ^c^	88.8 ± 1.8 ^b^

The IC_50_ value (μM) is defined as the concentration of 50% radical scavenging or relative reducing power, calculated by linear regression analysis, and expressed as the mean ± SD (*n* = 3). The IC_50_ values with different superscripts (^a^, ^b^, ^c^, or ^d^) in the same row have statistically significant differences (*p* < 0.05). Trolox ((±)-6-hydroxyl-2,5,7,8-tetramethylchromane-2-carboxylic acid) is the positive control for the antioxidant assays. The dose–response curves are shown in Appendix A. PTIO^•^, 2-phenyl-4,4,5,5-tetramethylimidazoline-1-oxyl-3-oxide radical.

**Table 2 molecules-26-01092-t002:** The BDE values (kcal/mol) of Fer-1, piceatannol, and astringin in aqueous and methanol solutions.

	Fer-1		Piceatannol		Astringin	
Aqueous	C_3_-H	439.46	C_3_O-H	400.13	C_5_O-H	397.43
	C_4_-H	414.90	C_5_O-H	399.77	C_3′_O-H	394.17
	C_6_-H	423.64	C_3′_O-H	394.22	C_4′_O-H	388.20
	C_1′_-H	407.65	C_4′_O-H	388.40		
	C_2′_-H	410.22				
	C_1_N-H	400.41				
	C_2_N-H	420.52				
	C_1′’_-H	398.28				
	C_2′’_-H	406.04				
	C_3′’_-H	407.79				
	C_4′’_-H	405.66				
Methanol	C_3_-H	441.81	C_3_O-H	402.78	C_5_O-H	402.49
	C_4_-H	422.87	C_5_O-H	402.31	C_3′_O-H	396.66
	C_6_-H	425.97	C_3′_O-H	396.90	C_4′_O-H	390.64
	C_1′_-H	410.05	C_4′_O-H	390.27		
	C_2′_-H	415.68				
	C_1_N-H	403.77				
	C_2_N-H	406.60				
	C_1′’_-H	399.97				
	C_2′’_-H	408.54				
	C_3′’_-H	410.14				
	C_4′’_-H	408.01				

BDE, bond dissociation enthalpy. The atom numbering is detailed in Figure 4.

## Data Availability

Data is contained within the article or supplementary material. The data presented in this study are available in this manuscript.

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
