# Peer review of "Comparison of Ferroptosis-Inhibitory Mechanisms between Ferrostatin-1 and Dietary Stilbenes (Piceatannol and Astringin)"

_molecules, 2021, doi:10.3390/molecules26041092_

Round 1

Reviewer 1 Report

The study of Chen et al investigated the difference in ferroptosis-inhibitory mechanisms between Fer-1 and dietary phytophenols piceatannol and astringin. Their results showed that the ferroptosis-inhibitory and electron-transfer potential levels decreased in order Ferrostatin-1 >> piceatannol > astringin. However, quantum calculations showed that Fer-1 has a lower ionization potential than stilbenes investigated in their study. The C4’O-H groups in the two stilbenes exhibited the lowest bond disassociation enthalpies. In addition, all three substances were found to produce corresponding dimer peaks using UPLC-ESI-Q-TOF-MS analysis. The authors conclude that Fer-1 mainly depends on the electron transfer of aromatic N-atoms to construct a redox recycle while piceatannol and astringin preferentially donate hydrogen atoms at the 4’-OH position to mediate the antioxidant mechanism that inhibits ferroptosis. The results of this study showed that piceatannol and astringin  may be safer ferroptosis inhibitors than arylamines.

The study is well written in a clear and comprehensible way. Introduction provide a sufficient background for a reader who is not familiar with this type of research. The design of the study is correct and the results are communicated in an understandable way. The results in the tables presented in the main text are supplemented by data in Supplementary files. Certificates of test substances are also included. The images are of suitable quality and are clearly presented.

The conclusions are in agreement with the results obtained and give experimentally confirmed results which show that stilbenes piceatannol and astringin may be safer ferroptosis inhibitors than arylamines. The authors also suggested that the findings based on the study of the two dietary stilbenes can be generalized into the dietary phytophenols family

I have only two minor comments on the article.

1.The number of rats used for the study should be added in the Animals section.

2. Why were bmMSCs used as a model cells, are the obtained results relevant for other cell models as well?

Reviewer 2 Report

The manuscript titled “Comparison of Ferroptosis-inhibitory Mechanisms between 2 Ferrostatin-1 and Dietary Stilbenes (Piceatannol and Astringin)” presented by Chen and co-workers have reported a very interesting comparison between Ferrostatin-1 and dietary phytophenols as ferroptosis inhibitory.

The manuscript is well written and presented.

However, there are some points to be clarified (reported below), which should be revised before the publication of the manuscript.

  • Figure 1: The caption has included three compounds, but only two structures are reported in this Figure. Structure of Ferrostation-1 should be insert.

  • Line 62: “Fer-1 and the two stilbenes were evaluated using cellular assays under the same conditions”. Please, insert here some references, or briefly explain, or, in alternative, report the Section where you explain these conditions.

  • Line 76 and 79: Please, verify the word “increases” and “biomolecules”. The dash should be removed.

  • Line 239-254: Section 3.1 Animals, biological kits, and chemical. In this section the authors have reported all chemicals and kit writing just “was from….” or “were from”; however, some verbs should be added (e.g. were purchase, were obtained, etc…)

Round 2

Reviewer 1 Report

I have no more comments on the manuscript.

Author Response

Dear reviewer,

Thank you very much for the academic positive comments.

The English language and style however have been carefully checked and revised. These revisions (9 in total) are highlighted in the main text.

Thank you very much!

Yours sinerely, 

Xican Li

Reviewer 2 Report

The authors of the manuscript titled “Comparison of Ferroptosis-inhibitory Mechanisms between 2 Ferrostatin-1 and Dietary Stilbenes (Piceatannol and Astringin)” have reported all suggested corrections in the revised version.

The manuscript should be accetpted for pubblication in Molecules.

Author Response

Dear reviewer,

Thank you very much for the academic positive comments.

The English language and style however have been carefully checked and revised. These revisions (9 in total) are highlighted in the main text.

Thank you very much!

Yours sinerely, 

Xican Li

This manuscript is a resubmission of an earlier submission. The following is a list of the peer review reports and author responses from that submission.